# Immunomodulatory Activity of Extracellular Vesicles of Kimchi-Derived Lactic Acid Bacteria (*Leuconostoc mesenteroides*, *Latilactobacillus curvatus*, and *Lactiplantibacillus plantarum*)

**DOI:** 10.3390/foods11030313

**Published:** 2022-01-24

**Authors:** Sang-Hyun Kim, Ji Hee Lee, Eun Hae Kim, Martin J. T. Reaney, Youn Young Shim, Mi Ja Chung

**Affiliations:** 1Institute of Animal Medicine, College of Veterinary Medicine, Gyeongsang National University, Jinju 52828, Korea; vetmicro@gnu.ac.kr; 2Department of Food Science and Nutrition, College of Health Welfare, Gwangju University, Gwangju 61743, Korea; wkssm2010@nate.com (J.H.L.); kjr01056@naver.com (E.H.K.); 3Department of Plant Sciences, University of Saskatchewan, Saskatoon, SK S7N 5A8, Canada; martin.reaney@usask.ca; 4Prairie Tide Diversified Inc., Saskatoon, SK S7J 0R1, Canada; 5Saskatchewan Oilseed Joint Laboratory, Department of Food Science and Engineering, Jinan University, Guangzhou 510632, China; 6Department of Integrative Biotechnology, Biomedical Institute for Convergence at SKKU (BICS), Sungkyunkwan University, Suwon 16419, Korea

**Keywords:** extracellular vesicles, lactic acid bacteria, *Lactiplantibacillus*, *Leuconostoc*, Kimchi probiotic bacteria, immunomodulatory activity

## Abstract

Lactic acid bacteria present in Kimchi, such as *Leuconostoc mesenteroides* (Lm), *Latilactobacillus curvatus* (Lc), and *Lactiplantibacillus plantarum* (Lp) produce extracellular vesicles (ECVs) that modulate immune responses. The ECVs of probiotic Kimchi bacteria are abbreviated as LmV, LcV, and LpV. Treatment of macrophages (RAW264.7) with ECVs (LmV, LcV, and LpV) increased the production of nitric oxide (NO), tumor necrosis factor (TNF)-α, and interleukin-6 (IL-6). Immunostimulatory effects exerted on the RAW264.7 cells were stronger after treatments with LmV and LcV than with LpV. Treatment of mice with LcV (1 mg/kg, orally) induced splenocyte proliferation and subsequent production of both NO and cytokines (INF-γ, TNF-α, IL-4, and IL-10). Furthermore, pre-treatment of macrophages and microglial cells with ECVs prior to LPS stimulation significantly attenuated the production of NO and pro-inflammatory cytokines (TNF-α, IL-1β, and IL-6). Therefore, ECVs (LmV, LcV, and LpV) prevent inflammatory responses in the LPS-stimulated microglial cells by blocking the extracellular signal-regulated kinase (Erk) and p38 signaling pathways. These results showed that LmV, LcV, and LpV from Kimchi probiotic bacteria safely exert immunomodulatory effects.

## 1. Introduction

Extracellular vesicles (ECVs) are membrane-bound vesicles that are ~100 nm in diameter and contain proteins, RNAs, and lipids. Almost all types of cells release ECVs to the extracellular space. As the ECV can contain a diverse repertoire of molecular cargo, they can modulate immunity in both in vitro and in vivo assays. ECVs can play essential roles in cell-to-cell communication and modulate immune responses under normal and pathological conditions [1]. Exosomes affect both innate and adaptive immunity by altering immunological responses including T-cell activation, immune suppression, and anti-inflammation [2]. However, there are no reports of the immunomodulatory activity of ECVs secreted from Kimchi lactic acid bacteria (LAB) such as *Leuconostoc* (*Le.*) *mesenteroides, Latilactobacillus* (*Ll.*) *curvatus*, and *Lactiplantibacillus (Lp.) plantarum*. The secretion and immune-modulating activities of ECVs from LAB, including *Lp. plantarum,* were reported elsewhere [3,4]. Although there are no previous reports of ECVs derived from *Le. mesenteroides* and/or *Ll. curvatus,* it is believed that all living bacteria produce ECVs [5,6] and that ECVs from many organisms modulate immunity.

Kimchi (Korean fermented vegetables) is a traditional Korean food fermented with LAB. The population of bacteria and bacterial metabolites increases during Kimchi fermentation [7,8]. LAB and their products modulate immunological processes and, thereby, can reduce inflammation [9,10]. However, there is no study on whether the ECVs of *Le. mesenteroides, Ll. curvatus,* and *Lp. plantarum* (LmV, LcV, and LpV) directly modulate immune processes.

Macrophages play a multifunctional role in host defense, and they can limit the pathogenesis of infectious organisms and affect degenerative disease processes. Macrophages and splenocytes including T-lymphocytes, B-lymphocytes, macrophages, etc., produce various inflammatory mediators and pro-inflammatory cytokines such as nitric oxide (NO), tumor necrosis factor (TNF)-α, interleukin-1β (IL-1β), and IL-6 after activation [11,12]. Concanavalin A stimulates the proliferation of treated splenocytes (T cell mitogen), and the induced proliferation can be used as the primary model of T cell activation [13,14]. The microglia are the resident “macrophages” of the brain’s innate immune system. The activation of microglia can be early evidence of hypersensitivity due to injury, infection, or neurodegenerative diseases. Overproduction of pro-inflammatory mediators by microglia can lead to neurotoxicity and systemic inflammation, which can exacerbate or trigger neurological diseases. Microglial-mediated inflammation contributes to the progression of Alzheimer’s disease (AD) [15,16].

Lipopolysaccharide (LPS)-stimulated macrophages and microglia release excess NO and pro-inflammatory cytokines, such as IL-1β, IL-6, and TNF-α. These NO and cytokines play a pivotal role in eliciting pathogenic inflammatory responses [15,17]. Furthermore, signaling pathways that lead to the expression of mitogen-activated protein kinase (MAPK), such as c-jun N-terminal kinase (JNK), extracellular signal-regulated kinase (Erk), and p38 are the most important signaling pathways in inflammatory responses [18].

In this study, the involvement of LmV, LcV, and LpV in the modulation of isolated T cell immune responses in mice treated for 10 days with LcV was investigated. Over-production of pro-inflammatory mediators from macrophage and microglia can lead to inflammatory diseases [15,17]. Therefore, the anti-inflammatory effects of LmV, LcV, and LpV in LPS-stimulated macrophages and microglia and downstream inflammatory MAPK signaling pathways were also examined in LPS-stimulated microglia.

## 2. Materials and Methods

### 2.1. Culture of LAB and Preparation of ECVs

*Le. mesenteroides* (KCTC3530), *Ll. curvatus* (KCTC3767), and *Lp. plantarum* (KCTC3104) were obtained from the Korean Collection for Type Cultures (KCTC, Daejeon, Korea). The LAB stocks were cultured anaerobically in *Lactobacilli* de Man Rogosa and Sharpe (MRS) broth medium (Difco Laboratories, Detroit, MI, USA). Then, *Lactiplantibacillus* culture supernatants were harvested from each culture by centrifugation (5000× *g*, 20 min). The procedure for purification of ECVs from *Lactiplantibacillus* culture supernatants was similar to methods previously described by Dean et al. [19]. Briefly, the supernatant from each bacterium culture was passed through a 0.45 μm filter and then applied to a 0.22 μm membrane filter using a vacuum apparatus to ensure there were no residual cells in the supernatant fraction. The filtrate was then ultracentrifuged at 150,000× *g* for 2 h in a fixed-angle (70 Ti) rotor using an Optima XE-100 model (Beckman Coulter, Brea, CA, USA). The supernatant was decanted, and the ECV pellet was incubated overnight at 4 °C in phosphate buffered saline (PBS) to resuspend the ECVs. The resuspended ECV preparations were purified, if necessary, by a further step using sucrose gradient ultracentrifugation described elsewhere [20]. Subsequently, the ECVs were lyophilized overnight using an SVQ-120 centrifugal vacuum concentrator (Operon, Gimpo-si, Gyeonggi-do, Korea) operated at 8000× *g* and −120 °C.

### 2.2. Microglial and Macrophage Cell Cultures

The murine macrophage cell line RAW264.7 and mouse microglial cell line EOC-20 were obtained from the American Type Culture Collection (ATCC, Rockville, MD, USA). The RAW264.7 cells and EOC-20 microglia were maintained in cell culture flasks containing Dulbecco’s modified Eagle’s medium (DMEM) with 10% fetal bovine serum (FBS, Gibco BRL, Grand Island, NY, USA) and 1% penicillin–streptomycin (PEST, Gibco BRL). The cells were cultured in a humidified incubator (Thermo Fisher Scientific, Waltham, MA, USA) at 37 °C with 5% CO_2_. Cells were counted using a hemocytometer so that a fixed number of cells would be seeded in 6-, 24-, and 96-well plastic culture plates, depending on the experiment, and cultured in 24- or 96-well plastic culture plates at a density of 1 × 10^4^ cells per well, for a 3-(4,5-dimethylthiazol-2-yl)-2,5-diphenyl-2H-tetrazolium bromide (MTT) assay. For enzyme-linked immunosorbent assay (ELISA) and Western blot analysis, cells were cultured in 24- and 6-well plastic culture plates at a density of 1 × 10^5^ cells per well and grown until 60–80% confluent. Cells were kept in a serum-free medium for 2 h prior to analysis. Following each treatment, cells or the cell culture medium were harvested for analysis by Western blot and ELISA analysis. Cells in 24- or 96-well plastic culture plates were used in an MTT assay.

### 2.3. Animals, Diets and Experimental Protocol

Six-week-old male Balb/c mice (25–30 g) were obtained from Orient Bio Inc. (Seongnam, Korea). The mice were allowed to live in controlled environmental conditions (ambient temperature, 23 ± 1 °C; relative humidity, 50 ± 5%; 12-h light/dark cycle). The animals were freely allowed drinking water and a standard pelleted diet. All protocols for animal experiments were approved by the Animal Ethics Committee of World Institute of Kimchi (approval ID: WIKIM IACUC 201938). Mice were maintained under standard laboratory conditions for one week prior to the experiments and then divided into two groups of 6–8 mice. The control group received 0.2 mL water per mouse. The H-LcV solution (0.2 mL of 1 mg/kg, H-LcV group) and L-LcV solution (0.2 mL of 0.1 mg/kg, L-LcV) were orally administered every day for 10 days to the H-LcV and L-LcV groups, respectively.

### 2.4. Splenocyte Isolation

The male Balb/c mice were sacrificed by CO_2_ inhalation and their spleens removed aseptically and placed into a tube containing RPMI 1640 with 10% FBS and 1% PEST. Spleens from each group of treated mice were individually used for lymphocyte proliferation. Splenocytes were isolated from a mouse spleen. The spleens were minced and passed through a 40 μm nylon cell strainer (BD Biosciences, San Jose, CA, USA) to obtain a single cell suspension and washed with PBS. Erythrocytes were removed by using a red blood cell (RBC) lysis buffer (Sigma-Aldrich Co., St. Louis, MO, USA). The cells were obtained by centrifugation (800× *g*, 4 min). After removal of the supernatant, an RBC lysis buffer was added to the cell pellet and re-suspended by shaking the tube. The tubes were placed at room temperature (21–23 °C) where light was blocked for 15 min. Then, the tubes were centrifuged at 800× *g* for 4 min and the supernatant was removed. The cell pellet was re-suspended with PBS and collected by centrifugation (800× *g*, 4 min). After two washes, cells were suspended in an RPMI 1640 medium containing 10% FBS and 1% PEST. The hemocytometer was used for counting the cell number, and cell viability was determined by the trypan blue exclusion method.

### 2.5. Splenocyte, T-Cell and B-Cell Culture

Splenocytes were cultured at 5 × 10^6^ cells/well in 24-well plates in complete RPMI 1640 medium in the absence or presence of the mitogen ConA (5 μg/mL) [21] added as a T cell stimulant. The culture plates were incubated at 37 °C in a CO_2_ incubator for 48 h. The cell culture media were collected in 1.5 mL tubes. After centrifugation (9400× *g*, 10 min, 4 °C), the supernatant was obtained and used or assays of the levels of NO and cytokines using the respective ELISA kits (eBioscience, Inc., San Diego, CA, USA).

### 2.6. Cell Viability

Cell viability was estimated using the MTT assay as previously described [22]. The MTT solution (5 mL, 5 mg/mL) and 45 mL new RPMI medium were mixed. After removing the old medium, the new medium containing the MTT solution was added and samples were incubated continuously for 4 h. Formazan produced by living cells was dissolved by adding dimethyl sulfoxide and incubating samples for 30 min. The absorbance at 570 nm was determined using a microplate reader (AMR-100, Hangzhou Allsheng Instruments Co., Ltd., Hangzhou, China).

### 2.7. Determination of NO and Cytokines

NO production by RAW264.7 macrophage cells, splenocytes, or microglial cells was measured in supernatants using Griess reagent for nitrite determination [23]. Briefly, 150 μL of sample and 150 μL of Griess reagent (A:B = 1:1, A: 0.1% N-(1-naphthyl)ethylenediamine dihydrochloride in distilled water, B: 1% sulfanilamide in 5% phosphoric acid) was mixed and incubated for 10 min at room temperature. The absorbance at 540 nm was measured with a microplate reader (AMR-100, Hangzhou Allsheng Instruments Co., Ltd.). The nitrite concentrations were determined from a calibration curve [23]. A sodium nitrite (NaNO_2_) standard solution was prepared by dissolving dried 50 μg NaNO_2_ in 10 mL distilled water. The NO concentration was determined using a standard calibration curve of 0–5 μg/mL of NaNO_2_. The concentration of secreted cytokines in cell culture supernatants was assessed following the protocols provided by the ELISA kit manufacturers (eBioscience, Inc.).

### 2.8. Western Blot Analysis

The microglial cells were pretreated with LmV, LcV, or LpV (2.5 and 5.0 μg/mL) in a DMEM medium for 2 h. The medium was then removed and replaced with DMEM containing 0.5 μg/mL LPS for 30 min. The radioimmunoprecipitation assay (RIPA) lysis buffer (50 mM Tris, 150 mM NaCl, 2 mM EDTA: ethylenediamine tetraacetic acid (EDTA), 1% Triton X-100, 0.1% sodium dodecyl sulfate (SDS), pH 7.8) containing protease and phosphatase inhibitor cocktails (Roche Applied Science, Mannheim, Germany) was added to the cells after removal of the medium, and cell lysates were collected by scraping the cells and then centrifuged at 18,300× *g*, 4 °C for 15 min. The supernatants were used for Western blot analysis. The protein (20 μg) was loaded onto 12% SDS-polyacrylamide gel electrophoresis (PAGE) gels and then transferred to nitrocellulose blotting membranes (GE Healthcare Life Science, Freiburg, Germany). The membranes were incubated with the diluted primary antibodies, such as p-p38, p-38, p-Erk 1/2, Erk 1/2, and β-actin, overnight at 4 °C. The membranes were washed three times with PBS containing 0.05% (*v/v*) Tween 20. The membranes were incubated for 1 h 30 min at 25 °C with HRP-conjugated anti-rabbit or mouse IgG antibodies (the secondary antibody). The diluted primary antibodies (1:1000 dilution) and secondary antibodies (1:2000 dilution) were used. After washing, immunoreactive proteins were visualized using an AbSignal Western blotting detection reagent kit (AbClon, Inc., Seoul, Korea). The membrane was then exposed to an X-ray film. Protein band intensities were quantified by SigmaGel software v. 1.0 (Jandel Scientific, San Rafael, CA, USA). Equal protein loading was confirmed by the β-actin antibody.

### 2.9. Statistical Analysis

Data were presented as mean ± standard deviation (SD), and the average values were derived from three to eight values per experiment. Each experiment was repeated at least three times. All data were analyzed using the one-way analysis of variance (ANOVA) with IBM SPSS for Windows version 18.0 software (SPSS Inc., Chicago, IL, USA). The differences in the results between treatments were confirmed by Duncan’s multiple range test. Statistical significance was accepted at *p* < 0.05.

## 3. Results and Discussion

### 3.1. Effects of ECVs on Macrophage Viability, NO Production and Secreted Cytokine

The effects of LmV, LcV, and LpV on RAW264.7 macrophage viability were determined. RAW264.7 cells were incubated with LmV, LcV, and LpV (0.1–10.0 μg/mL), respectively, for 24 h. Cell viability of the RAW264.7 macrophages was not altered by LmV, LcV, and LpV (Figure 1A). These concentrations were not cytotoxic from 0.5 μg/mL to 5.0 μg/mL. Thus, these concentrations were used for subsequent experiments. As immune response parameters, the production of NO and cytokines such as TNF-α and IL-6 by LmV, LcV, and LpV were evaluated in RAW264.7 cells. The LmV, LcV, and LpV treatments induced significant NO production in RAW264.7 macrophages. Among the ECVs tested, LcV increased NO production more than LmV and LpV (Figure 1B).

NO is an important signal in the regulation of immune response and host defense against pathogenic bacteria, viruses, and tumor cells. Therefore, a NO assay was used as a convenient tool for detection of immune response [24]. NO production of RAW264.7 cells in response to *Lactobacillus*
*brevis* KCCM12203P and *Lp. plantarum* 200655 was reported to be 21.83 and 11.38 μM [25,26]. Lower NO production was reported in RAW264.7 exposed to *Lp. plantarum* extracts compared to extracts from other *Lactiplantibacillus* strains [25,26], which also indicated that cells exposed to LpV produced less NO than cells treated with LmV and LcV. There are many studies on the immunomodulatory effects of LAB from Kimchi [25,26,27,28]. Still, there are no studies on the immunomodulatory effects of LmV, LcV, and LpV in RAW264.7 macrophages, microglial cells, primary cells (splenocyte, T-cells and B-cells), and Balb/c mice models.

The effects of LmV, LcV, and LpV on TNF-α and IL-6 secretion in RAW264.7 cells were measured by ELISA. LmV, LcV, and LpV significantly increased TNF-α production compared with controls. LmV and LcV induced a dose-dependent increase in IL-6 secretion by RAW264.7 cells while treatment with LpV at 2.5 and 5.0 μg/mL concentration had no effect (Figure 2B). Treatments with LmV, LcV, and LpV induced TNF-α and IL-6 production. Among the three ECVs (LmV, LcV, and LpV), LmV stimulated the greatest increase of TNF-α and IL-6 protein. LmV, LcV, and LpV might promote RAW264.7 cells’ immune responses. Among ECVs isolated from the *Lactiplantibacillus* stains, LpV was less effective than LcV in terms of the immunostimulatory effects of RAW264.7 macrophages. Subsequently, studies in animal models were conducted to determine if in vitro results could be reproduced.

In the current study, immunostimulatory effects of LmV, LcV, and LpV on RAW264.7 cells were evaluated. LAB was used to improve immunity, and heat-killed probiotics, as well as live probiotics, are reported to have immune-stimulating effects [21]. LAB isolated from Kimchi increased levels of NO, TNF-α, and IL-6 [28]. *Lp. plantarum* 200655 isolated from Kimchi showed immune-enhancing effects [25] and heat-killed *Lp. plantarum* Ln1 isolated from Kimchi exert their immune-stimulating effects through NO and pro-inflammatory cytokine (TNF-α, IL-1β, and IL-6) production in macrophages [29]. Like Jang’s research results [29], LpV exerts immune-stimulating effects through production of NO, TNF-α, IL-1β, and IL-6. However, there was no significant change in the NO, IL-1β, and IL-6 levels at 10^7^ CFU/mL *Lp. plantarum* Ln1 compared to the controls [29]. In this study, the IL-6 content of cells treated with 2.5 μg/mL LpV were not different to the control group, but IL-6 content of cells treated with 2.5 μg/mL LmV and 2.5 μg/mL LcV were significantly increased. When the RAW264.7 cells were exposed to LpV, LmV, and LcV at 5.0 μg/mL, cells significantly increased production of NO, TNF-α, IL-1β, and IL-6. Therefore, it was found that even small amounts of LpV, LmV, and LcV have strong immune-enhancing effects. LpV was excluded from animal experiments as it had the lowest immunostimulatory effects of RAW264.7 macrophages among ECVs. In this study, LcV was used in animal experiments because *Ll. curvatus* is a major lactic acid bacterium in Kimchi [30] and has biological effects [31,32] as well as a high immunostimulatory effects of LcV. *Ll. curvatus* was responsible for much of the Kimchi fermentation [30] and Jo et al. [31] reported that the *Ll. curvatus* WiKim38 isolated from baechu (Chinese cabbage) Kimchi induced cytokine production in bone marrow-derived dendritic cells. In our previous study, *Ll. curvatus* inhibited N-nitrosodimethylamine formation and directly degraded *N*-nitrosodimethylamine [32].

### 3.2. Cell Proliferation and Change of NO of ConA-stimulated Splenocytes Isolated from Mouse Treated with LcV

We further investigated spleen lymphocyte proliferation as this leads to the activation of lymphocytes and cell-mediated humoral immune responses. The previous studies indicated that administering red ginseng extract [33] or sulfated polysaccharide from a green seaweed [34] to mice once a day for 10 or 14 days enhanced the immune-modulating activities of LAB ECVs including those from *Lp. plantarum* [3,4]. In the Balb/c mouse model, the L-LcV was orally administered for 10 days (once/day). Mice were treated with oral LcV for 10 days (once/day) at concentrations of 0.1 mg/kg body weight (BW) (L-LcV) and 1 mg/kg BW (H-LcV). Splenocytes were isolated from the spleens of mice that had been given LcV. The effect of LcV on ConA-stimulated splenocyte proliferation was determined by exposure of the mice to the mitogen ConA. Administration of H-LcV (1.0 mg/kg BW) significantly enhanced ConA-stimulated splenocyte proliferation. Splenocytes isolated from mice treated with L-LcV (0.1 mg/kg BW) at a low dose showed no significant differences in ConA-stimulated splenocyte proliferation when compared with control groups (Figure 3A). The effect of LcV on mitogen-driven lymphocyte NO production of T cells is presented in Figure 3B. Mice treated with the low dose of L-LcV (0.1 mg/kg BW) showed no significant change in T cell NO production (Figure 3B). However, mice treated with the high dose of H-LcV (1 mg/kg BW) showed significantly increased T cell NO production (Figure 3B). The NO concentration of splenocytes in the H-LcV group, without mitogen treatment, also increased significantly compared to control splenocytes.

Spleen cell proliferation of T lymphocytes, B lymphocytes, and macrophages after immunostimulation is critical for activation of cell-mediated and humoral immune responses [35,36]. Con A was used as the common mitogen for the activation of T lymphocytes [37]. The cell viability of T lymphocytes was increased by LmV administration compared with control groups, but there were no differences in viability among groups in splenocytes. This suggests that cell viability may be dependent on immune cell type.

### 3.3. Cytokines Production in ConA-Stimulated Splenocytes Isolated from Mouse Treated for 10 Days with LcV

To understand the modulatory effects of LcV on splenocyte Th1/Th2 immune response after ConA stimulation., ELISA was used to quantify cytokine production (INF-γ and TNF-α for Th1 and IL-4 and IL-10 for Th2). As shown in Figure 4, the results indicate that administration of H-LcV (1 mg/kg BW) increased INF-γ, TNF-α, IL-4, and IL-10 production in ConA-stimulated splenocytes, but there were no significant increases in the production of INF-γ, TNF-α, IL-4, and IL-10 after administration of L-LcV (0.1 mg/kg BW). The production of INF-γ, TNF-α, IL-4, and IL-10 from the splenocytes without mitogen in the H-LcV group also increased significantly compared to the control group (Figure 4). These results indicated that H-LcV modulated the Th1 and Th2 responses via upregulation of INF-γ, TNF-α, IL-4, and IL-10 in T cells and splenocytes.

The two classes of T helper (Th) cells, Type 1 (Th1) and Type 2 (Th2) cells, regulate immune responses [38]. Th1 cells produce interferon-gamma (INF-γ), IL-2, IL-12, and TNF-α, which are responsible for phagocyte-dependent protective responses. Th2 cells produce IL-4, IL-5, IL-6, IL-10, and IL-13, which are responsible for antibody production and B cell proliferation [39,40]. Th1 cells and Th2 cells regulate the immune response by balancing each other. Kang et al. [41] reported that cytokine productions by both Th1 and Th2 were increased by tuna cooking juice concentrate. Similarly, our study showed that the production of INF-γ, TNF-α, IL-4, and IL-10 increased in splenocytes and T cells isolated from mice treated for 10 days with 1 mg/kg BW LcV.

In our laboratory, LmV had a similar immune-enhancing effect at a concentration 200 times lower than that of the water extract of a mixture from bellflower (*Platycodon grandiflorum*), deodeok (*Codonopsis lanceolata*), corn silk (*Zea mays*), and sweet potato peel (*Ipomoea batatas*) in a ratio of 1.0: 1.0: 0.5: 3.0 (*w*/*w*) under the same in vitro experimental conditions [21,23]. Therefore, in this study, 100 times and 1000 times less than the plant mixture [21] was administered in animal experiments under the same conditions. Unfortunately, Balb/c mice given 0.1 mg L-LcV/kg BW had no change in immune activity (Figure 3 and Figure 4).

The excessive release of pro-inflammatory cytokines can cause inflammation which is associated with various diseases including cancer, diabetes, and Alzheimer’s disease. Thus, the immunomodulatory and immunostimulatory compounds are drug candidates for the regulation of the immune system and protection from disease. Therefore, the inhibition of inflammatory responses under excessive inflammatory conditions by LmV, LcV, and LpV was studied.

### 3.4. The Anti-Inflammatory Effect of ECVs against Excessive NO and Cytokines Production in LPS-Stimulated RAW264.7 Cells

The concentration of NO was determined in RAW264.7 macrophage cells treated with LPS (0.1, 0.5, 1.0, 2.5, and 5.0 μg/mL). All ECVs (LmV, LcV, and LpV) significantly decreased LPS-stimulated NO production (Figure 5A). Moreover, LmV, LcV, and LpV also significantly inhibited production of pro-inflammatory cytokines, such as TNF-α, IL-1β, and IL-6 (Figure 5B–D). However, production of NO, TNF-α, IL-1β, and IL-6 were increased 6.1–8.0, 40.8–46.2, 3.7–4.1, and 64.0–67.9 fold, respectively, by treatment with 2.5 μg/mL LmV-, LcV-, or LpV-pretreatment higher in LPS-stimulated RAW264.7 cells than those in the control group (1-fold). The exopolysaccharides produced by *Le. mesenteroides* isolated from Korean Kimchi exhibited anti-inflammatory effects in LPS-stimulated RAW264.7 cells [42]. *Ll. curvatus* WiKim38 isolated from Kimchi induced IL-10 production in dendritic cells, and oral administration of *Ll. curvatus* WiKim38 increased survival and decreased inflammation in mice treated with dextran sodium sulfate to induce colitis [31]. The acidic exopolysaccharide (EPS103) from *Lp. plantarum* JLAU103 promoted the release of NO, TNF-α, and IL-6 in RAW264.7 macrophages and EPS103 reduced excessive release of NO, TNF-α, and IL-6 in LPS-stimulated RAW264.7 cells [43]. Although there are no research reports describing the effects of ECVs on inflammation, results demonstrating the immunomodulatory effects of Kimchi LAB-derived substances support the results of this study.

### 3.5. The Anti-Inflammatory Effect of ECVs against Excessive NO and Cytokines Production in LPS-Stimulated Microglial Cells

Parkinson’s, Alzheimer’s, and other neurodegenerative diseases are induced by neuroinflammation of brain cells including microglia, which are brain-resident myeloid cells that support the central nervous system. Activated microglia represent a common pathological feature of several neurodegenerative diseases, including Alzheimer’s disease [16]. NO secretion in LPS-induced microglial cells were inhibited by LmV-, LcV-, or LpV-pretreatment (Figure 6A). LmV, LcV, and LpV inhibited LPS-induced TNF-α, IL-1β, and IL-6 production (Figure 6B–D).

### 3.6. The Anti-Inflammatory Effect of ECVs on MAPKs Signaling Pathways of LPS-Stimulated Microglial Cells

The effects of LmV, LcV, and LpV on LPS-stimulated microglial cells were determined by Western blot analysis. Protein expression levels indicated LPS-induced phosphorylation of MAPKs (p38, Erk, and JNK) pathways in microglial cells. Figure 7 shows that pretreatments with different ECVs (LmV, LcV, and LpV) at two concentrations (2.5 and 5 μg/mL) prior to LPS treatment could significantly decrease the activation of phosphorylated Erk and p38 protein compared to LPS-treated cells. LPS showed strong p-Erk and p-p38 protein expression levels compared to the controls. LmV, LcV, and LpV did not affect phosphorylation of JNK in LPS-stimulated microglial cells (data not shown). This anti-inflammatory activity on microglial cells is mediated by effects on p38 and Erk signaling pathways. 

Inflammatory mediator production in activated microglia can be inhibited by natural bioactive compounds like cordycepin, which is isolated from *Cordyceps militaris.* Production of NO and pro-inflammatory cytokines is significantly inhibited by cordycepin suppression of MAPK signaling pathways in LPS-stimulated microglia [44]. The results of studies by Jeong et al. [44] are similar to those of this study. It is therefore concluded that LmV, LcV, and LpV might prove helpful in treating neurodegenerative diseases such as Parkinson’s, Alzheimer’s, and other diseases through the anti-neuroinflammatory effects in activated microglia. LmV, LcV and LpV have the potential for use as immunomodulatory functional foods.

## 4. Conclusions

RAW264.7 macrophage cells and T-lymphocytes isolated from the spleen of mice exhibited immunostimulatory effects when administered with LcV. Specifically, LcV induced production of NO and pro-inflammatory cytokines. Moreover, LmV, LcV, and LpV reduced NO, TNF-α, IL-1β, and IL-6 release from LPS-stimulated RAW264.7 macrophages. Finally, LmV, LcV, and LpV significantly inhibited NO and pro-inflammatory cytokine production by LPS-stimulated microglia. This inhibitory effect on LPS-stimulated inflammatory mediator production by microglia is associated with the suppression of the MAPK signaling pathways through phosphorylation of p38 and Erk. These findings suggest that LmV, LcV, and LpV possess potent dual immunomodulatory activities and they may be considered as candidates for the development of functional foods and medicines that act by regulation of immunomodulatory activity.

## Figures and Tables

**Figure 1 foods-11-00313-f001:**
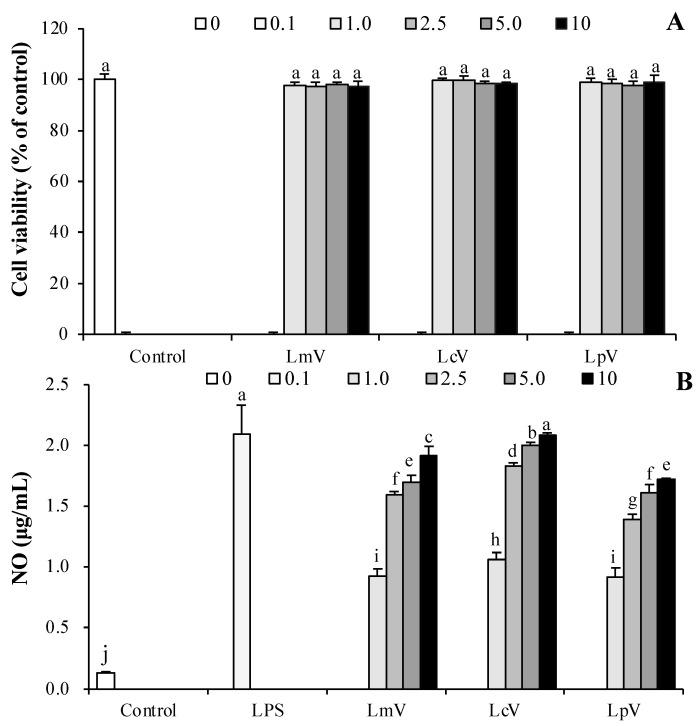
Effect of *Le. mesenteroides, Ll. curvatus* or *Lp. plantarum*-derived ECVs (LmV, LcV, or LpV) on (**A**) cell viability and (**B**) NO production of RAW264.7 macrophage cells. The cells were treated with concentrations (0.1, 0.1, 1.0, 2.5, 5.0 and 10 μg/mL) of LmV, LcV, and LpV, respectively, for 24 h. Values are expressed as the mean ± SD (*n* = 4) and those followed by different letters (a–j) within a property are significantly different (*p* < 0.05) according to Duncan’s multiple range test.

**Figure 2 foods-11-00313-f002:**
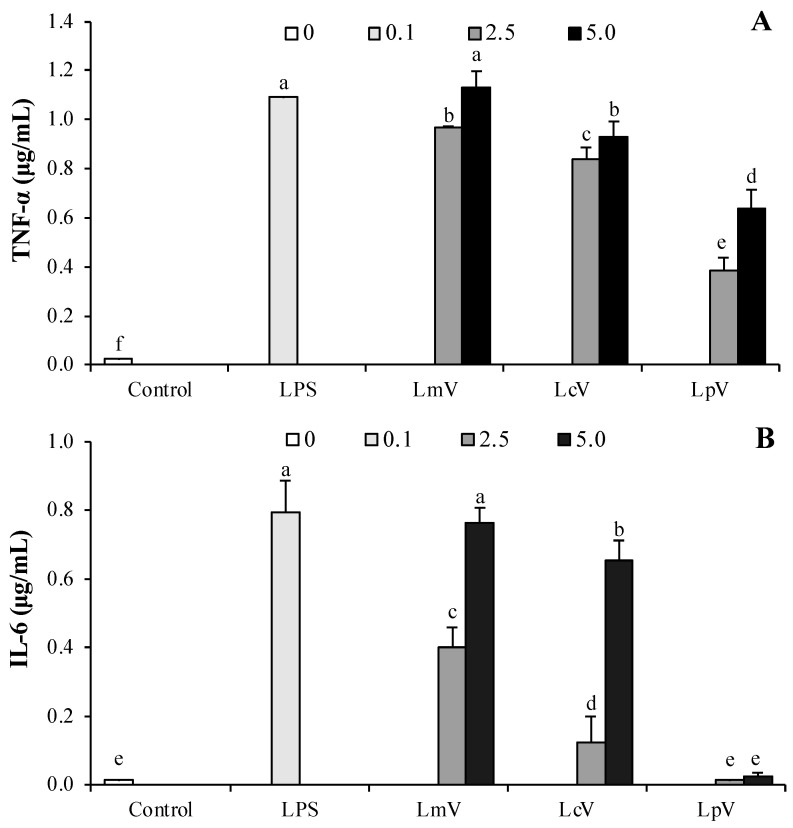
Stimulatory effect of *Le. mesenteroides*, *Ll. curvatus,* or *Lp. plantarum*-derived ECVs (LmV, LcV, or LpV) on the production of (**A**) TNF-α and (**B**) IL-6 in RAW264.7 macrophages. LPS (0.1 μg/mL) was used as a positive control. Values are expressed as the mean ± SD (*n* = 4) and those followed by different letters (a–f) within a property are significantly different (*p* < 0.05) according to Duncan’s multiple range test.

**Figure 3 foods-11-00313-f003:**
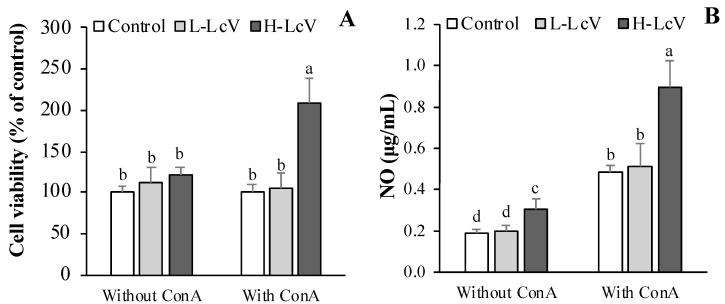
Splenocyte (**A**) proliferation and (**B**) NO production of mice administered with *Ll. curvatus*-derived ECVs (LcV) for 10 days. LcV orally treated for 10 days (once/day) at the concentration of 0.1 mg/kg BW (L-LcV) and 1 mg/kg BW (H-LcV). Isolated splenocyte were stimulated with mitogen (ConA) for 24 h or 48 h. Values are expressed as the mean ± SD (*n* = 6) and followed by different letters (a–d) within a property are significantly different (*p* < 0.05) according to Duncan’s multiple range test.

**Figure 4 foods-11-00313-f004:**
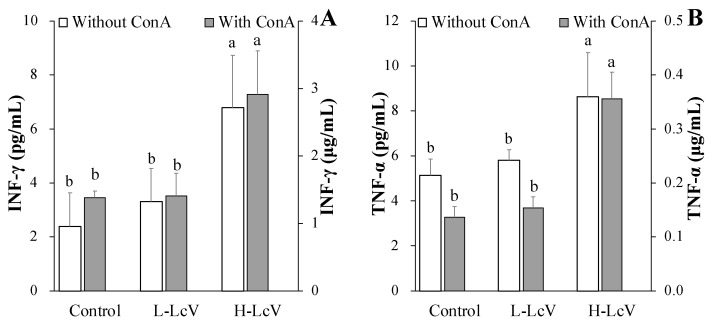
Splenocyte (**A**) INF-γ, (**B**) TNF-α, (**C**) IL-4, and (**D**) IL-10 of mice administered with *Ll. curvatus*-derived ECVs (LcV) for 10 days. LcV orally treated for 10 days (once/day) at the concentrations of 0.1 mg/kg BW (L-LcV) and 1 mg/kg BW (H-LcV). Isolated splenocyte were stimulated with mitogen (ConA or LPS) for 24 h or 48 h. Values are expressed as the mean ± SD (*n* = 6) and those followed by different letters (a,b) within a property are significantly different (*p* < 0.05) according to Duncan’s multiple range test.

**Figure 5 foods-11-00313-f005:**
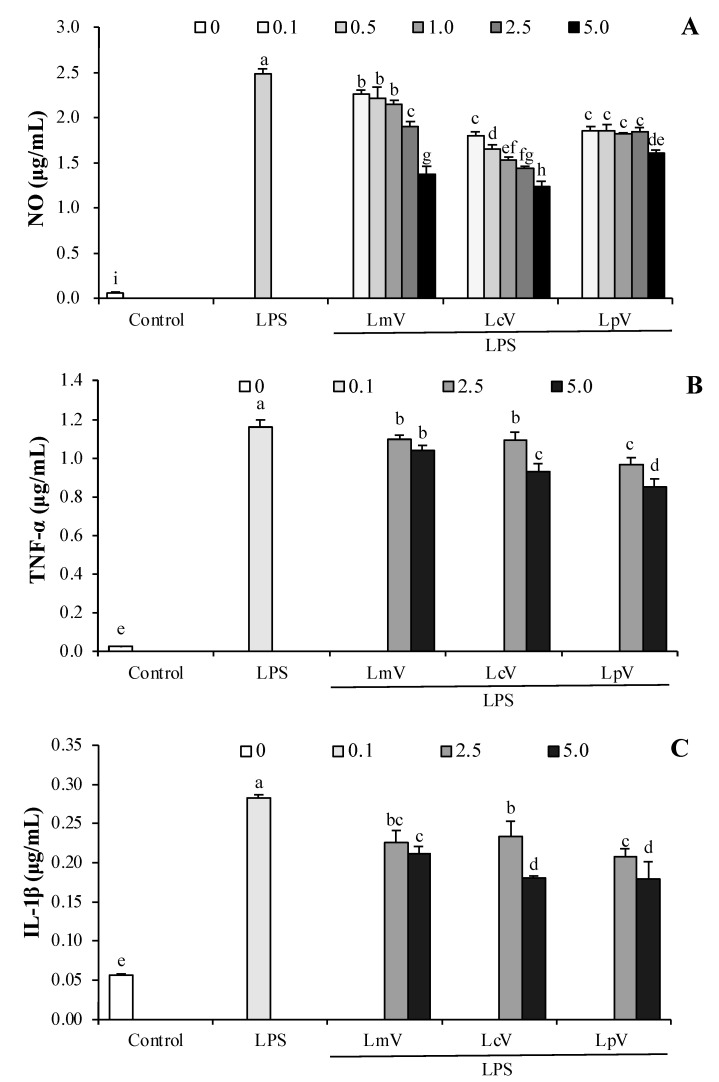
Effect of *Le. mesenteroides, Ll. curvatus,* or *Lp. plantarum*-derived ECVs (LmV, LcV, or LpV) on (**A**) the production of NO, (**B**) TNF-α, (**C**) IL-1β, and (**D**) IL-6 in LPS-stimulated RAW264.7 macrophages. RAW264.7 macrophages were treated with LmV, LcV, or LpV for 2 h prior to the addition of LPS (0.5 μg/mL), and the cells were further incubated for 24 h. Values are expressed as the mean ± SD (*n* = 4) and those followed by different letters (a–i) within a property are significantly different (*p* < 0.05) according to Duncan’s multiple range test.

**Figure 6 foods-11-00313-f006:**
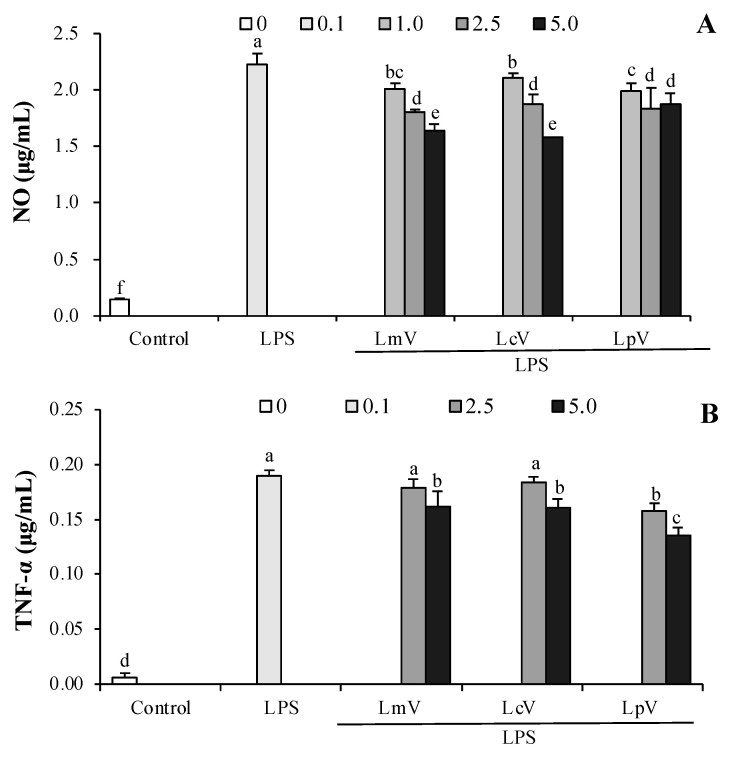
Effect of *Le. Mesenteroides, Ll. curvatus,* or *Lp. plantarum*-derived ECVs (LmV, LcV, or LpV)) on (**A**) the production of NO, (**B**) TNF-α, (**C**) IL-1β, and (**D**) IL-6 in LPS-stimulated microglial cells. Microglial cells were treated with LmV, LcV or LpV for 2 h prior to the addition of LPS (0.5 μg/mL), and the cells were further incubated for 24 h. Values are expressed as the mean ± SD (*n* = 4) and those followed by different letters (a–f) within a property are significantly different (*p* < 0.05) according to Duncan’s multiple range test.

**Figure 7 foods-11-00313-f007:**
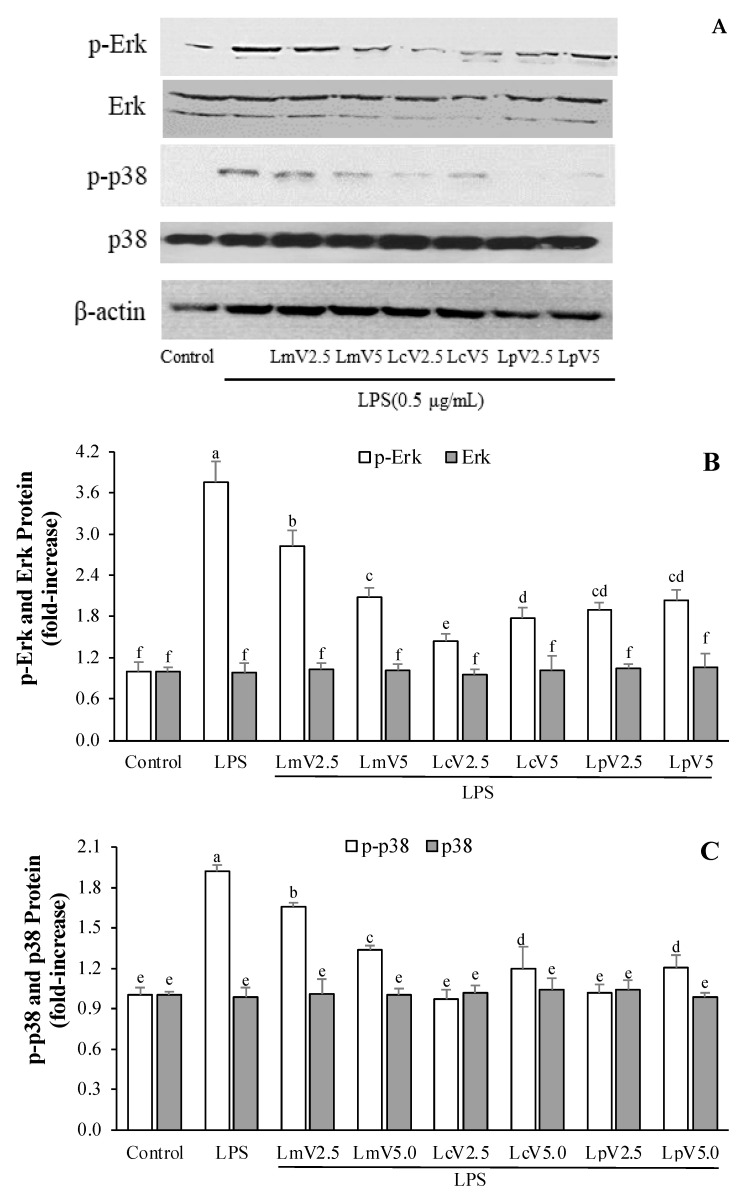
Effect of *Le. mesenteroides, Ll. curvatus,* or *Lp. plantarum*-derived ECVs (LmV, LcV, or LpV) on LPS-induced phosphorylation of Erk and p38 in microglia (**A**–**C**). The cells were pretreated with the LmV, LcV, or LpV (2.5 and 5.0 μg/mL) for 2 h, followed by treatment with the LPS (0.5 μg/mL) for 30 min. The Erk, p-Erk, p38, and p-p38 levels in each sample were normalized to the β-actin levels. Control and LPS were control cells (LPS and sample-untreated cells) and LPS group cells (cell treated with the LPS only), respectively. The phosphorylation of Erk and p38 was abbreviated by p-Erk and p-p38. Values are expressed as the mean ± SD (*n* = 4) and those followed by different letters (a–f) within a property are significantly different (*p* < 0.05) according to Duncan’s multiple range test.

## Data Availability

The data of the current study are available from the corresponding author on reasonable request.

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
