# Peer review of "Immunomodulatory Activity of Extracellular Vesicles of Kimchi-Derived Lactic Acid Bacteria (Leuconostoc mesenteroides, Latilactobacillus curvatus, and Lactiplantibacillus plantarum)"

_foods, 2022, doi:10.3390/foods11030313_

Round 1
Reviewer 1 Report
This manuscript is deserved for its publication in Foods with minor corrections.
- Lines 72-74: The sentence – "In this study, we demonstrated that LmV, LcV, and LpV were involved in modulating the isolated murine macrophage and T-cell immune responses in mice treated for 10 73 days with LcV." - sounds like a conclusion rather than a formulation of the research problem or purpose of the paper. Please verify.
- The Authors use the abbreviations EVs and ECVs interchangeably in the manuscript. So do both refer to the production of extracellular vesicles by lactic acid bacteria present in Kimchi?
- Lines 95-96: Were all purified EVs lyophilised by the Authors? Please provide details of the lyophilisation equipment used?
- The Authors administered LcV only to mice and did not administer LmV and LpV. Why? This information should appear in the manuscript.
- What substance was used to prepare the calibration curve for the determination of NO? Please indicate the concentration range in which this substance has been used.
- Lines 196-198: The Authors reported that RAW264.7 cells were incubated with LmV, LcV and LpV in the concentration range of 0.5-5.0 µg/mL. In contrast, Figure 1 gives concentrations from 0.1 to 10 µg/mL. Please explain.
- Lines 310-311: What was "the plant mixture extract"?
Author Response
Manuscript ID foods-1518478 entitled “Immunomodulatory Activity of Extracellular Vesicles of Kimchi-derived Lactic Acid Bacteria (Leuconostoc mesenteroides, Lactobacillus curvatus, and Lactobacillus plantarum)”
Thank you for your patience. We have revised our manuscript (Manuscript ID: foods-1518478) according to the reviewers’ comments/suggestions and would like to thank all reviewers for their critical feedback in making this manuscript more polished. We have listed Reviewer 1’s comments and answered them in sequence. In addition to the changes made to the manuscript as recommended by the reviewers, we have also tried to improve the manuscript’s structure. We appreciate the reviewers’ thoughtful comments and critiques and hope this response addresses the overall quality of this manuscript for publication.
Responses to Reviewer 1 Comments and Suggestions:
This manuscript is deserved for its publication in Foods with minor corrections.
- Lines 72-74: The sentence – "In this study, we demonstrated that LmV, LcV, and LpV were involved in modulating the isolated murine macrophage and T-cell immune responses in mice treated for 10 73 days with LcV." - sounds like a conclusion rather than a formulation of the research problem or purpose of the paper. Please verify.
Response: We corrected the sentence (Lines 76-78).
- The Authors use the abbreviations EVs and ECVs interchangeably in the manuscript. So do both refer to the production of extracellular vesicles by lactic acid bacteria present in Kimchi?
Response: EVs and ECVs refer to the production of extracellular vesicles by bacteria including lactic acid bacteria present in Kimchi. ECVs and EVs are the same abbreviations, so we unified them as ECV in the manuscript.
- Lines 95-96: Were all purified EVs lyophilised by the Authors? Please provide details of the lyophilisation equipment used?
Response: ECVs were frozen by the author. We provided details of the lyophilisation equipment used in the materials and methods of the revised manuscript and are as follows:
Subsequently, all purified ECVs were stored at –20 °C or lyophilized in a freeze-dryer (SVQ-120, Operon, Gimpo-si, Gyeonggi-do, Korea) for further use (Lines 100, 101).
- The Authors administered LcV only to mice and did not administer LmV and LpV. Why? This information should appear in the manuscript.
Response: The following sentences have been added to the revised manuscript, and relevant references have been inserted in the Results, Discussion, and References sections (lines 261–270).
LpV was excluded from animal experiments as it had the lowest immunostimulatory effects of RAW264.7 macrophages among ECVs. In this study, LcV was used in animal experiments because L. curvatus is a major lactic acid bacterium in kimchi (Lee et al., 2005) and imparts of biological effects (Jo et al. 2016; Kim et al., 2017) as well as a high immunostimulatory effect of LcV in this study. L. curvatus was responsible for much of kimchi fermentation (Lee et al., 2005) and Jo et al. (2016) reported that the L. curvatus WiKim38 isolated from baechu (Chinese cabbage) kimchi induced cytokine production in bone marrow-derived dendritic cells. In our previous study, L. curvatus inhibited N-nitrosodimethylamine formation and directly degraded N-nitrosodimethylamine (Kim et al., 2017).
Lee, J.S.; Heo, G.Y.; Lee, J.W.; Oh, Y.J.; Park, J.A.; Park, Y.H.; Pyun, Y.R.; Ahn, J.S. Analysis of kimchi microflora using denaturing gradient gel electrophoresis. Int. J. Food Microbiol. 2005, 102, 143–150.
Jo, S.G.; Noh, E.J.; Lee, J.Y.; Kim, G.; Choi, J.H.; Lee, M.E.; Song, J.H.; Chang, J.Y.; Park, J.H. Lactobacillus curvatus WiKim38 isolated from kimchi induces IL-10 production in dendritic cells and alleviates DSS-induced colitis in mice. J. Microbiol. 2016, 54, 503–509.
Kim, S.H.; Kang, K.H.; Kim, S.H.; Lee, S.H.; Lee, S.H.; Ha, E.S.; Sung, N.J.; Kim, J.G.; Chung, M.J. Lactic acid bacteria directly degrade N-nitrosodimethylamine and increase the nitrite-scavenging ability in kimchi. Food Control. 2017, 71, 101–109.
- What substance was used to prepare the calibration curve for the determination of NO? Please indicate the concentration range in which this substance has been used.
Response: We used sodium nitrite (NaNO2) to prepare the calibration curve to determine. NO. We indicated the concentration range in which this substance has been used as follows:
A sodium nitrite (NaNO2) standard solution was prepared by dissolving dried 50 μg NaNO2 in 10 mL distilled water. NO concentration was determined using a standard calibration curve of 0–5 μg/mL of NaNO2.
- Lines 196-198: The Authors reported that RAW264.7 cells were incubated with LmV, LcV and LpV in the concentration range of 0.5-5.0 µg/mL. In contrast, Figure 1 gives concentrations from 0.1 to 10 µg/mL. Please explain.
Response: RAW264.7 cells were incubated with LmV, LcV and LpV in the concentration range of 0.1–10.0 µg/mL. Thus, LmV, LcV and LpV (0.5–5.0 µg/mL) was corrected to LmV, LcV and LpV (0.1–10.0 µg/mL)
- Lines 310-311: What was "the plant mixture extract"?
Response: The plant mixture extract was water extract of a mixture from bellflower (Platycodon grandiflorum), deodeok (Codonopsis lanceolata), corn silk (Zea mays), and sweet potato peel (Ipomoea batatas) in a ratio of 1.0: 1.0: 0.5: 3.0 (w/w). The following sentences were added to the revised manuscript.
In our laboratory, LmV had a similar immune-enhancing effect at a concentration 200 times lower than that of water extract of a mixture from bellflower (Platycodon grandiflorum), deodeok (Codonopsis lanceolata), corn silk (Zea mays), and sweet potato peel (Ipomoea batatas) in a ratio of 1.0: 1.0: 0.5: 3.0 (w/w) under the same in vitro experimental conditions [21,23].

Reviewer 2 Report
The paper titled ‘Immunomodulatory Activity of Extracellular Vesicles of Kimchi-derived Lactic Acid Bacteria (Leuconostoc mesenteroides, Lactobacillus curvatus, and Lactobacillus plantarum)’ aimed to reveal the immunomodulation and anti-inflammation effects of three LAB obtained from Kimchi, while the RAW 264.7 macrophages, murine splenocytes, and mouse microglial cell were used as the cell models. That’s such an interesting result that the three LAB chosen in this work could exhibit similar bioactivities.
However, there are still many questions about this article:
There was no associated information shown in this paper about if the bioactivities of LAB were altered after the digestion. Thus, an in vitro digestion experience is strongly suggested.
The chemical components, that might exhibit these bioactivities of these samples, were not clarified.
Some bacteria were considered able to release the EVs to modulate the immune responses in vitro and in vivo. However, is there any evidence that these selected LAB (Leuconostoc mesenteroides, Lactobacillus curvatus, and Lactobacillus plantarum) could modulate the immune responses through the EVs secretion? Please confirm and cite some related references in the introduction section.
The phenomenon, LAB could combine with LPS, has been mentioned by many researchers, do the authors believes that the LAB chosen in this work exhibited the anti-inflammation effects in the same way or others?
Why the mice were orally administered for 10 d but not for other periods?
The units used in this paper were inconsistent (e.g., g and rpm).
The ELISA kit was used to assess the contents of cytokines, but the fold, or times, were chosen by the authors. This is neither common nor suitable.
The English should be improved. There are some grammatical and spelling mistakes in the article.
Based on the points mentioned above, my suggestion is the paper needs a major revision.
Author Response
Manuscript ID foods-1518478 entitled “Immunomodulatory Activity of Extracellular Vesicles of Kimchi-derived Lactic Acid Bacteria (Leuconostoc mesenteroides, Lactobacillus curvatus, and Lactobacillus plantarum)”
Thank you for your patience. We have revised our manuscript (Manuscript ID: foods-1518478) according to the reviewers’ comments/suggestions and would like to thank all reviewers for their critical feedback in making this manuscript more polished. We have listed Reviewer 2’s comments and answered them in sequence. In addition to the changes made to the manuscript as recommended by the reviewers, we have also tried to improve the manuscript’s structure. We appreciate the reviewers’ thoughtful comments and critiques and hope this response addresses the overall quality of this manuscript for publication.
Responses to Reviewer 2 Comments and Suggestions:
The paper titled ‘Immunomodulatory Activity of Extracellular Vesicles of Kimchi-derived Lactic Acid Bacteria (Leuconostoc mesenteroides, Lactobacillus curvatus, and Lactobacillus plantarum)’ aimed to reveal the immunomodulation and anti-inflammation effects of three LAB obtained from Kimchi, while the RAW 264.7 macrophages, murine splenocytes, and mouse microglial cell were used as the cell models. That’s such an interesting result that the three LAB chosen in this work could exhibit similar bioactivities.
However, there are still many questions about this article:
There was no associated information shown in this paper about if the bioactivities of LAB were altered after the digestion. Thus, an in vitro digestion experience is strongly suggested.
Response: Our hypothesis is that the extracellular vesicles produced by lactobacilli can exert anti-inflammatory effects and modulate immunity. The first line of the introduction states “Extracellular vesicles (ECVs) are membrane-bound vesicles that are ~100 nm in diameter that contain proteins, RNAs, and lipids.” This statement broadly describes these structures, and this might help the reviewer understand the nature of ECVs. That said, a detailed analysis of ECV structure, the digestion of ECVs and the relationship of each ECV constituent to bioactivity goes beyond the answer to our hypothesis. We thank the reviewer for this observation as it is a promising area for future research.
The chemical components, that might exhibit these bioactivities of these samples, were not clarified.
Response: The chemical components of Lactobacillus-derived membrane vesicles (MV) and bioactivities of these MV-conveyed components were characterized and reported previously (Dean et al., 2019: Champagne-Jorgensen et al, 2012).
Dean, S.N.; Leary, D.H.; Sullivan, C.J.; Oh, E.; Walper, S.A. Isolation and characterization of Lactobacillus-derived membrane vesicles. Sci. Rep. 2019, 9(1), 877. doi: 10.1038/s41598-018-37120-6.
Champagne-Jorgensen, K.; Mian, M.F.; McVey Neufeld, K.A.; Stanisz, A.M.; Bienenstock, J. Membrane vesicles of Lacticaseibacillus rhamnosus JB-1 contain immunomodulatory lipoteichoic acid and are endocytosed by intestinal epithelial cells. Sci. Rep. 2021, 11(1), 13756. doi: 10.1038/s41598-021-93311-8.
Some bacteria were considered able to release the EVs to modulate the immune responses in vitro and in vivo. However, is there any evidence that these selected LAB (Leuconostoc mesenteroides, Lactobacillus curvatus, and Lactobacillus plantarum) could modulate the immune responses through the EVs secretion? Please confirm and cite some related references in the introduction section.
Response: The following sentences and references were inserted in the introduction part. “Secretion and immune-modulating activity of ECVs from LAB, including L. plantarum were reported elsewhere (Li et al., 2017; Hao et al., 2021). Although there are no previous reports of ECVs derived from Le. mesenteroides and/or L. curvatus, it is believed that all living bacteria produce ECVs (Ñahui Palomino et al., 2021; Nishiyama et al., 2021) and that ECVs from many organisms modulate immunity.”
- Li, M.; Lee, K.; Hsu, M.; Nau, G.; Mylonakis, E.; Ramratnam, B. Lactobacillus-derived extracellular vesicles enhance host immune responses against vancomycin-resistant enterococci. BMC Microbiol. 2017, 17(1), 66. doi: 10.1186/s12866-017-0977-7.
- Hao, H.; Zhang, X.; Tong, L.; Liu, Q.; Liang, X.; Bu, Y.; Gong, P.; Liu, T.; Zhang, L.; Xia, Y.; Ai, L.; Yi, H. Effect of extracellular vesicles derived from Lactobacillus plantarumQ7 on gut microbiota and ulcerative colitis in mice. Front Immunol. 2021, 12, 777147. doi: 10.3389/fimmu.2021.777147.
- Ñahui, Palomino, R.A.; Vanpouille, C.; Costantini, P.E.; Margolis, L. Microbiota-host communications: Bacterial extracellular vesicles as a common language. PLoS Pathog. 2021, 17(5), e1009508. doi: 10.1371/journal.ppat.1009508.
- Nishiyama, K.; Yokoi, T.; Sugiyama, M.; Osawa, R.; Mukai, T.; Okada, N. Roles of the cell surface architecture of Bacteroides and Bifidobacterium in the gut colonization. Front Microbiol. 2021, 12, 754819. doi: 10.3389/fmicb.2021.754819.
The phenomenon, LAB could combine with LPS, has been mentioned by many researchers, do the authors believes that the LAB chosen in this work exhibited the anti-inflammation effects in the same way or others?
Response: Hu et al., (2021) reported immune suppression mediated by the ECVs on LPS-induced inflammatory responses in ex vivo experiments. The ECVs inhibited Th1- and Th17-mediated inflammatory responses and enhanced immunoregulatory cell-mediated immunosuppression in splenic lymphocytes of LPS-challenged chickens through the activation of macrophages. Choi et al. (2020) reported that ECVs reduced the expression of the LPS-induced pro-inflammatory cytokines IL-1α, IL-1β, IL-2, and TNFα and increased the expression of the anti-inflammatory cytokines IL-10 and TGFβ in in vitro experiments. Accordingly, we believe that the ECVs of the LAB used in this study could show similar anti-inflammatory effects (Hu et al., 2021; Choi et al., 2020).
Hu, R.; Lin, H.; Wang, M.; Zhao, Y.; Liu, H.; Min, Y.; Yang, X.; Gao, Y, Yang M. Lactobacillus reuteri-derived extracellular vesicles maintain intestinal immune homeostasis against lipopolysaccharide-induced inflammatory responses in broilers. J Anim Sci Biotechnol. 2021, 12(1), 25. doi: 10.1186/s40104-020-00532-4.
Choi, J.H., Moon, C.M., Shin, TS. et al. Lactobacillus paracasei-derived extracellular vesicles attenuate the intestinal inflammatory response by augmenting the endoplasmic reticulum stress pathway. Exp Mol Med. 2020, 52, 423–437, https://doi.org/10.1038/s12276-019-0359-3
Why the mice were orally administered for 10 d but not for other periods?
Response: The following sentences have been added to the revised manuscript, and relevant references have been inserted in the Results and Discussion (lines 274-278), and References sections.
The previous studies indicated that administering red ginseng extract (Hyun et al., 2018) or sulfated polysaccharide from a green seaweed (Kim et al. 2016) to mice once a day for 10 or 14 days enhanced the immune-modulating activities of LAB ECVs including those from L. plantarum (Li et al., 2017; Hao et al., 2021). In the Balb/c mouse model, the L-LcV was orally administered for 10 days (once/day) in this study.
Hyun, S.H.; Kim, Y.S.; Lee, J.W.; Han, C.K.; Seon, P.M.; So, S.H. Immunomodulatory effects of arginine-fructose-glucose enriched extracts of red ginseng. J. Korean Soc. Food Sci. Nutr. 2018, 47, 1–6.
Kim, J.K.; Park, J.H.; Jang, E.H.; Surayot, U.; You, S.G. Immunomodulatory effect of sulfated polysaccharides and its low molecular form isolated from Enteromorpha prolifera in BALB/c mice. J. Chitin Chitosan 2016, 21, 82–88.
The units used in this paper were inconsistent (e.g., g and rpm).
Response: We unified them as g in the manuscript.
The ELISA kit was used to assess the contents of cytokines, but the fold, or times, were chosen by the authors. This is neither common nor suitable.
Response: Cytokines and NO concentration was determined using a standard calibration curve. The contents of cytokines were changed, and the NO content was also changed to μg/mL and the figures were drawn again.
The English should be improved. There are some grammatical and spelling mistakes in the article.
Response: English grammar and spelling errors have been corrected according to the reviewer's comments.
Based on the points mentioned above, my suggestion is the paper needs a major revision.

Reviewer 3 Report
In the paper coauthored by Kim and coll., immunomodulatory activity of extracellular vesicles from Kimchi lactic acid bacteria were fully examined. The findings in this manuscript are beneficial for understanding the function of LAB from Kimchi production. The paper needs to be thoroughly checked and revised. The following comments need to be addressed.
- The English of this manuscript need to be improved.
- As the authors indicated there are numerous studies evaluating the immunomodulatory activity of LAB from Kimchi (such as reference 19-20), thus the novelty of the present study must be expressed clearly.
- Line 48-49, a reference is needed after “The population of bacteria, bacterial metabolites and ECV increase during kimchi fermentation”.
- Line 220-221, the authors indicate that “LmV, LcV and LpV significantly increased TNF-α and IL-6 production compared with controls”. It was contradictory with the expression in Line 221-222. From Fig. 2, it was also exhibited that LpV at 2.5 and 5.0 μg/mL had no effect on IL-6 production.
- Line 250, “CoA-stimulated” should revised as “ConA-stimulated”.
- Line 290, “L-LcV modulated the Th1 and Th2 response” should revised as “H-LcV modulated the Th1 and Th2 response”.
- Line 300, a reference is needed after “Type 1 (Th1) cells and Type 2 (Th2), regulate 300 immune responses”.
Author Response
Manuscript ID foods-1518478 entitled “Immunomodulatory Activity of Extracellular Vesicles of Kimchi-derived Lactic Acid Bacteria (Leuconostoc mesenteroides, Lactobacillus curvatus, and Lactobacillus plantarum)”
Thank you for your patience. We have revised our manuscript (Manuscript ID: foods-1518478) according to the reviewers’ comments/suggestions and would like to thank all reviewers for their critical feedback in making this manuscript more polished. We have listed Reviewer3’s comments and answered them in sequence. In addition to the changes made to the manuscript as recommended by the reviewers, we have also tried to improve the manuscript’s structure. We appreciate the reviewers’ thoughtful comments and critiques and hope this response addresses the overall quality of this manuscript for publication.
Responses to Reviewer 3 Comments and Suggestions:
In the paper coauthored by Kim and coll., immunomodulatory activity of extracellular vesicles from Kimchi lactic acid bacteria were fully examined. The findings in this manuscript are beneficial for understanding the function of LAB from Kimchi production. The paper needs to be thoroughly checked and revised. The following comments need to be addressed.
The English of this manuscript need to be improved.
Response: English grammar and spelling errors have been corrected according to the reviewer's comments.
As the authors indicated there are numerous studies evaluating the immunomodulatory activity of LAB from Kimchi (such as reference 19-20), thus the novelty of the present study must be expressed clearly.
Response: The following sentences have been added to the revised manuscript, and relevant references have been inserted in the Results and Discussion (lines 227–230), and References sections.
There are many studies on the immunomodulatory effects of LAB from Kimchi (25–28). Still, there are no studies on the immunomodulatory effects of LmV, LcV, and LpV in Raw264.7 macrophages, microglial cells, primary cells (splenocyte, T-cells and B-cells) and Balb/c mice models.
Line 48-49, a reference is needed after “The population of bacteria, bacterial metabolites and ECV increase during kimchi fermentation”.
Response: The references are added after “The population of bacteria and bacterial metabolites increase during kimchi fermentation” ECV is deleted from the sentence. The following sentences have been added to the revised manuscript, and relevant references have been inserted in the Results, Discussion, and References sections.
The population of bacteria and bacterial metabolites increases during kimchi fermentation (Lee et al., 2012; Choi et al., 2019).
Lee, E.H.; Lee, M.J.; Song, Y.O. Comparison of fermentation properties of winter kimchi stored for 6 months in a kimchi refrigerator under ripening mode or storage mode. J. Korean Soc. Food Sci. Nutr. 2012, 41, 1619–1625.
Choi, Y.J.; Yong, S.J.; Lee, M.J.; Park, S.J.; Yun, Y.R.; Park, S.H.; Lee, M.A. Changes in volatile and non-volatile compounds of model kimchi through fermentation by lactic acid bacteria. LWT. 2019, 105, 118–126.
Line 220-221, the authors indicate that “LmV, LcV and LpV significantly increased TNF-α and IL-6 production compared with controls”. It was contradictory with the expression in Line 221-222. From Fig. 2, it was also exhibited that LpV at 2.5 and 5.0 μg/mL had no effect on IL-6 production.
Response: “and IL-6” of line 232 were removed. The following sentences were added to the revised manuscript.
LmV, LcV, and LpV significantly increased TNF-α production compared with control.
Line 250, “CoA-stimulated” should revised as “ConA-stimulated”.
Response: Two places of lines 311 and 322 were corrected.
Line 290, “L-LcV modulated the Th1 and Th2 response” should revised as “H-LcV modulated the Th1 and Th2 response”.
Response: Corrected (Line 315).
Line 300, a reference is needed after “Type 1 (Th1) cells and Type 2 (Th2), regulate immune responses”.
Response: The following sentences have been added to the revised manuscript, and relevant references have been inserted in the Results, Discussion, and References sections.
Type 1 (Th1) cells and Type 2 (Th2), regulate immune responses (Rengarajan et al., 2000).
Rengarajan, J.; Szabo, S.J.; Glimcher, L.H. Transcriptional regulation of Th1/Th2 polarization. Immunol. Today 2000, 21, 479–483.

Round 2
Reviewer 2 Report
I agree paper acceptance.
Author Response
Thank you for your patience. We have revised our manuscript (Manuscript ID: foods-1518478) according to the reviewers’ comments/suggestions and would like to thank all reviewers for their critical feedback in making this manuscript more polished. We have listed Academic Editor’s comments and answered them in sequence on Jan 6, 2022.
Reviewer 3 Report
The manuscript has been revised, thus it can be accepted.
Author Response
Thank you for your patience. We have revised our manuscript (Manuscript ID: foods-1518478) according to the reviewers’ comments/suggestions and would like to thank all reviewers for their critical feedback in making this manuscript more polished. We have listed Academic Editor’s comments and answered them in sequence on Jan 6, 2022.
This manuscript is a resubmission of an earlier submission. The following is a list of the peer review reports and author responses from that submission.